



**Observational benchmarks inform representation of soil organic carbon dynamics in land surface models**

Kamal Nyaupane[1], Umakant Mishra[2*], Feng Tao[3], Kyongmin Yeo[4], William J. Riley[5], Forrest M. Hoffman[6] and Sagar Gautam[2]

[1]Environmental science and Engineering Program, The University of Texas at El Paso, El Paso, TX 79968, United States.

[2] Biomaterials & Biomanufacturing, Sandia National Laboratories, Livermore, CA, 94550, United States.

[3]Ministry of Education Key Laboratory for Earth System Modelling, Department of Earth System Science, Tsinghua University, Beijing, 100084, China

[4]IBM Thomas J. Watson Research Center, Yorktown Heights, NY, 10562, United States.

[5]Earth & Environmental Sciences, Lawrence Berkeley National Laboratory, Berkeley, CA, 94720, United States.

[6]Climate Change Institute, Oak Ridge National Laboratory, Oak Ridge, TN, 37830, United States.

*Corresponding author Email: umishra@sandia.gov





**Abstract**
Representing soil organic carbon (SOC) dynamics in Earth system models (ESMs) is a key
source of uncertainty in predicting carbon climate feedbacks. Machine learning models can help
identify dominant environmental controllers and their functional relationships with SOC stocks.
The resulting knowledge can be implemented in ESMs to reduce uncertainty and better predict
SOC dynamics over space and time. In this study, we used a large number of SOC field
observations (n = 54,000), geospatial datasets of environmental factors (n = 46), and two
machine learning approaches (Random Forest (RF) and Generalized Additive Modeling (GAM))
to: (1) identify dominant environmental controllers of global and biome-specific SOC stocks, (2)
derive functional relationships between environmental controllers and SOC stocks, and (3)
compare the identified environmental controllers and predictive relationships with those in
Coupled Model Intercomparison Project phase six (CMIP6) models. Our results showed that
diurnal temperature, drought index, cation exchange capacity, and precipitation were important
observed environmental controllers of SOC stocks. RF model predictions of global-scale SOC
stocks were relatively accurate ($R^2 = 0.61$, RMSE = 0.46 kg m$^{-2}$). In contrast, precipitation,
temperature, and net primary productivity explained >96% of ESM-modeled SOC stock
variability. We also found very different functional relationships between environmental factors
and SOC stocks in observations and ESMs. SOC predictions in ESMs may be improved
significantly by including additional environmental controls (e.g., cation exchange capacity) and
representing the functional relationships of environmental controllers consistent with
observations.



**Keywords**: Environmental controllers, Earth system models, soil organic carbon, net primary

productivity, machine learning, model benchmarking

**1. Introduction**

Soil is the largest actively cycling carbon pool in terrestrial ecosystems and stores almost twice

the amount of carbon as in the current atmosphere (Lal, 2016). A small change in soil carbon

stocks can lead to large changes in the atmospheric $CO_2$ concentration and future climate change

trajectories. Soils also play a crucial role in sequestering atmospheric $CO_2$ as soil organic carbon

(SOC) (Hinge et al., 2018). Thus, sequestration, protection, and sustainable management of SOC

stocks can be a promising climate mitigation strategy (Lal, 2020). Accurate representation of

global SOC storage and its environmental controllers are essential for predicting realistic

changes of SOC under different land use and climate change scenarios. Yet, no consensus exists

among current Earth system models (ESMs) in representing the spatial distributions of global

SOC storage and its fate under future climate change scenarios (Friedlingstein et al., 2014.;

Arora et al., 2020).

Multiple environmental variables, including climatic and topographic factors, land use history,

and edaphic properties, have been identified as possible controllers of SOC storage (Georgiou et

al., 2021; Mishra et al., 2022). Current ESMs, however, use the effects of only a limited number

of environmental factors in representing SOC storage and dynamics. A recent study that

compared SOC stocks from multiple ESMs against observations indicated a large knowledge gap

in both ESMs and observations (Georgiou et al., 2021). Therefore, it is important to compare

ESM simulations against global SOC observational datasets to evaluate model performance and

identify key environmental controllers of global SOC storage.





Benchmarking ESM simulations with observed data is a common approach for model evaluation
(Luo et al., 2012; Todd-Brown et al., 2013; Collier et al., 2018). Through comparing model
simulations with observations, model strengths, deficiencies, and needed improvements can be
identified. The resulting understanding from SOC benchmarking could lead to new ESM land
model structures (by identifying key processes) and new parameterizations (by quantifying key
relationships between SOC and environmental variables). Thus, benchmarking analysis of ESMs
is an effective tool to reduce uncertainties in predicting SOC dynamics and can provide more
realistic information for managing SOC under changing climate conditions (Lauer et al., 2017).
Currently ESMs predict SOC stocks primarily with model representations that depend on soil
temperature, moisture, and belowground net primary production (Todd-Brown et al., 2013).
ESMs capture the positive correlation between NPP and precipitation, resulting in high SOC
stocks for areas with high NPP in moist regions (Sun et al., 2016). Higher temperature increases
soil respiration, which, in the short-term, reduces SOC storage. In the longer-term, increased soil
respiration can release nutrients, leading to increased plant growth, belowground carbon inputs,
and thereby SOC stocks; the balance of these factors can take centuries to manifest (Mekonnen
et al., 2022). Soil respiration temperature sensitivity is often defined based on $Q_{10}$ or Arrhenius
equations in ESMs (Wynn et al., 2006), although low- and high-temperature modifications to
these relationships are likely needed (Jiang et al., 2013; Azizi-Rad et al., 2022).
In a previous U.S. continental-scale study, we derived empirical non-linear relationships between
SOC and environmental factors that produced comparable prediction accuracy to a random forest
(RF) machine learning approach (Mishra et al., 2022). We apply a similar approach in this study
in both global field observations and ESMs to (1) identify key observed environmental controllers
of, and functional relationships with, global SOC stocks and (2) evaluate ESMs with these





observational benchmarks. Simulated SOC stocks from three CMIP6 ESMs (i.e., Community
Earth System Model (CESM, Hurrell et al., 2013); U.K. Earth System Model (UKESM, Sellar et
al., 2019); Beijing Climate Center model (BCC, Xiao-Ge et al., 2019) were benchmarked with
50,000 SOC profile observations across the globe. We used a machine learning (i.e., random
forest) approach with 46 environmental factors to identify the key environmental controllers of
SOC stocks at the global scale. We then applied a generalized additive model (GAM) to derive the
predictive relationships between these key environmental factors and SOC stocks in observations
and ESM simulations.   Specific objectives of this study were to: (1) identify dominant
environmental controllers of SOC stocks in field observations and CMIP6 ESMs, (2) derive
observed and ESM-modeled functional relationships between environmental factors and SOC
stocks, and (3) analyze these functional relationships to inform needed improvements in ESM
representations of SOC dynamics.
**2. Materials and Methods**
*2.1 Soil organic carbon stock observations*
We used two datasets of SOC stocks for the topsoil layer (i.e., 0 – 30 cm) and the whole soil profile
(i.e., 0 – 100 cm). The World Soil Information Service (WoSIS) compiled SOC profiles across the
globe after quality assessment. The 2019 snapshot of the WoSIS dataset contained 111,380 soil
profiles with SOC content information (unit: g C g-soil$^{-1}$) at different soil depths (Batjes et al.,
2020). We estimated the SOC stock (g C m$^{-2}$) at different soil layers using:
$$SOC\ Stock = SOC\ Content \times \left(1 - \frac{G}{100}\right) \times BD \times D \qquad (1)$$
where G is the coarse fragment fraction (%); BD is the bulk density of soil (g m$^{-3}$); and D is the
soil layer depth (m).





When the measured bulk density value was absent from the dataset, we used a pedo-transfer
function (Yigini et al., 2018) to estimate the soil bulk density:
$BD = \alpha + \beta \times exp(-\gamma \times OM)$                    (2)
Where OM is organic matter, equivalent to SOC×1.724, with SOC content in percent (%); $\alpha$, $\beta$,
and $\gamma$ are fitting parameters. We found $\alpha = 0.32$, $\beta = 1.30$, and $\gamma = 0.0089$ after fitting WoSIS data
to this equation.
Another dataset we used in this study was compiled from Mishra et al. (2021). This dataset
contained 2,546 soil profiles with SOC stock (g C m$^{-3}$) information from permafrost regions in
North America, northern Eurasia, and the Qinghai-Tibet Plateau. In total, we used 113,926 soil
profile observations from these two data sources. SOC stocks of different soil layers were then
summed to SOC stocks in 0 – 30 cm and 0 – 100 cm depth intervals. Because not all these soil
profiles covered the whole 0 – 30 cm or 0 – 100 cm intervals, we used a total of 54,000 soil profiles
that included SOC stock information for both depth intervals. The geographical distributions of
soil profiles used in this study are shown in Figure 1. Because SOC stock values across the globe
were highly skewed, we used a natural logarithm transformation in this study.
*2.2 Environmental predictors of SOC stocks*
The storage and cycling of SOC are controlled by multiple environmental factors. In this study,
we used observations of 46 environmental variables, which represented major soil forming factors
(McBratney et al., 2003.). Twenty-one of the 46 environmental variables were climatic variables,
including annual average temperature, precipitation, evapotranspiration, drought severity index,
and statistics for different temporal scales (e.g., during the wettest and driest quarter in a year).
Thirteen of the 46 variables described soil properties (e.g., clay content, sand content, silt content,
soil texture, pH, and cation exchange capacity). Six variables represented topographic factors (e.g.,



elevation and soil depth). Six variables represented land use and land cover types. All the
categorical variables were converted to integer variables and the environmental variables were
resampled to a common 1 km resolution. The environmental factors, their original spatial
resolution, and data sources are provided in the supporting information (Table S1).

***2.3 Selection of dominant environmental controllers of SOC stocks***

We used RF to select dominant environmental predictors of SOC stocks within biomes and at
global scale in both observations and ESMs. RF is an ensemble learning method, which is an
extension of the classical Classification and Regression Trees (CART). Building a collection of
uncorrelated CARTs through bootstrapping the samples and applying the random subspace method
at each branch of the trees, RF improves the prediction performance (Breiman, 2001; Wiesmeier
et al., 2011; Mishra et al., 2020). RF is well known for its strength in modeling highly nonlinear
relationships between the predictors and is robust to overfitting (Chagas et al., 2016). Moreover,
RF is not very sensitive to the choice of the hyperparameters, which makes RF one of the most
popular off-the-shelf model for many classification and regression problems.
In this study, we trained the RF model using SOC content as a response variable and environmental
factors as predictors. The model performance was evaluated using the coefficient of determination
($R^2$) and root mean square error (RMSE). A 10-fold cross-validation was used to compute $R^2$ and
RMSE. Biome-specific analyses were conducted on a subset of the global dataset. For biome
classification, we used the IGBP land classes (Loveland and Belward, 1997). The "Random-
Forest" package in R was used to train a RF model using all the observed environmental factors in
the dataset and to identify dominant environmental controllers of SOC stocks. Prior to fitting into
the final model, we performed a potential collinearity test among the environmental variables by





calculating pairwise correlations and variance influence factors. Predictors showing a variance
influence factor (VIF) value greater than 10 were omitted, leaving 14 uncorrelated environmental
predictors of SOC stocks in the observations.

**_2.4 Generalized additive model_**

Generalized additive model (GAM) is an extension of generalized linear models, which employs
spline functions to model nonlinear relationships between predictor and response variables (Arnold
et al., 2013). In GAM, the relationship between predictor and response variable can be modeled as
(Hastie and Tibshirani, 1987):
$$Y = C + \sum_{i=1}^{p} f_i(X_i) \tag{3}$$
Here, Y is the response variable (SOC), $C$ is a constant, $X_i$ are the environmental controller
variables, $f_i$ is a spline function for $X_i$, and p is the total number of environmental controllers. We
used the "mgcv" package in R to build GAMs for the observations as well as CMIP6 ESMs
(Arnold et al., 2013). The performance of GAMs was evaluated by using $R^2$ and RMSE.
**_2.5 Earth system model outputs_**
We downloaded and aggregated the SOC and environmental controller data from three ESMs that
participated in CMIP6: Community Earth System Model (Hurrell et al., 2013.), U.K. Earth System
Model (Sellar et al., 2019), and Beijing Climate Center model (Xiao-Ge et al., 2019). These ESMs
included most of the environmental factors used by CMIP6 ESMs. ESMs did not report depth-
dependent soil carbon projections, making direct comparison with depth-dependent SOC
observations difficult. The majority of land models used in ESMs were designed to simulate topsoil



carbon for topsoil depth; thus, we assumed that the simulated soil carbon is contained within 1 m
of soil profile to simplify comparison with observations.

**3.**     **Results**

***3.1***     ***Descriptive statistics of SOC observations***

The average global SOC stock in the 0 - 1 m depth interval was 13.5 kg C m$^{-2}$, ranging from 0.14-
435.3 kg C m$^{-2}$. Summary statistics of SOC stocks at global scale and within different biomes is
presented in Table 1. The standard deviation showed a similar spread in SOC stock values in
croplands (n=21820), savannas (n=9807) and grasslands (n=5938). However, in forests (n=12164)
and shrublands (n=3769), the standard deviation was higher indicating a large range in SOC stock
values. Distributions of total SOC stocks in different biomes are presented in Figure 2. Across
different biomes, forests contain the largest organic carbon content globally, with a mean value of
15.9 kg C m$^{-2}$ and standard deviation 20.7 kg C m$^{-2}$.

***3.2***     ***Dominant environmental controllers of SOC stocks in observations and ESMs***

At the global scale, we found that diurnal temperature, drought severity index, annual
temperature, and cation exchange capacity are the dominant environmental controllers of SOC
stocks in observations (Figure 3). By including all the environmental controllers, the RF model
explained 61% of observed global spatial SOC variation. $R^2$ ranged from 48% in savannas to
65% in croplands (Table 2) and the importance of key environmental controllers varied between
biomes (Figure 4). In croplands, precipitation, drought, diurnal temperature, and cation exchange
capacity were identified as the dominant controllers of SOC stocks. In grasslands, annual
temperature, cation exchange capacity, and sand content were the dominant controllers. In
forests, cation exchange capacity, precipitation, and temperature were dominant controllers. In



shrublands, annual temperature, soil pH, and cation exchange capacity were the most important
controllers. In savannas, soil related variables, temperature, and precipitation were the most
important controllers. Across all land cover types, we found that cation exchange capacity and
seasonal climatic variables were the dominant environmental controllers of SOC stocks.
In contrast, the RF model with 8 environmental variable predictors made near-perfect
predictions of ESM simulated SOC stocks (average $R^2 = 0.95$, $R^2$ values for UKESM, CESM,
and BCC model were 0.99, 0.89, and 0.98, respectively). In contrast to the results obtained from
the observed SOC stocks, the dominant controllers of ESM simulated SOC stocks were annual
temperature, net primary productivity (NPP), and annual precipitation (Figure 5). In particular,
NPP was by far the most dominant predictor of SOC stocks in the UKESM.

*3.2 Predictive relationships between environmental factors and SOC stocks*
Dominant environmental controllers of observed SOC stocks identified by the RF model
were used in GAM to derive predictive relationships. We retrieved explicit analytical
expressions by fitting the splines derived from GAM in the observation dataset. Notwithstanding
its role as the sole carbon source to soil, our results did not show NPP as a strong controller on
observed SOC stocks (Figure 6a). In contrast with field observations, all ESMs showed
significant dependence (exponential increase) of SOC stocks on NPP. Our results also showed
that observed SOC stocks increased almost linearly with observed annual precipitation (Figure
6b). In contrast, ESMs show different relationships between SOC and precipitation. We found a
nonlinearly increasing SOC with precipitation in CESM, an initial sharply increasing and then
decreasing relationship in UKESM, and a decreasing relationship in BCC ESM. On the
relationship between SOC storage and soil texture and elevation, ESMs do not capture the



observed relationships. Our results indicated that observed SOC stocks decreased with clay
content in the interval between 0 and 20%, and then increased with clay content above 20%
(Figure 6c). Observed SOC stocks increased with silt content up to 55% and then decreased
(Figure 6d).

SOC stock functional relationships differed between the three ESMs and in many cases

differed with the relationships we derived from observations. In terms of the effects of annual
temperature on modeled SOC storage, we found that SOC stocks decreased with annual
temperature and were most sensitive to temperature in the range between 0 and 10$^{o}$C (Figure 6e).
However, while the three ESMs captured the general negative relationship between SOC storage
and temperature, none of them correctly described the varying sensitivity of SOC in different
temperature ranges (especially in extreme temperature ranges <0$^{o}$C and >20$^{o}$C). In representing
the control of elevation on SOC storage, only UKESM showed consistent patterns with
observations, where SOC storage remained stable when the elevation was lower than 2000 m and
decreased when the elevation was higher than 2000 m (Figure 6f).

**Discussion**
Previous studies have suggested that the spatial variation of SOC is dependent on multiple
environmental factors such as climatic and edaphic variables, geography, and vegetation. Here,
we found that climatic variables (i.e., temperature and precipitation) are the most important
controllers of global SOC stocks, followed by edaphic variables (i.e., cation exchange capacity),
topography (i.e., elevation), and vegetation (i.e., NPP). Using boosted regression trees, Luo et al.
(2021) studied edaphic and climatic controls on SOC dynamics at different soil depths and found
that soil type and climatic variables are the most important variables in explaining the SOC





stocks (Luo et al., 2021). In this study, we found that seasonal climatic variables such as diurnal
temperature range and precipitation seasonality are among the most important environmental
controllers in explaining the spatial variation of SOC stocks. This result indicates the critical role
of seasonal and interannual climatic variables in understanding SOC dynamics.

The importance of climatic variables on global SOC storage emerges from close links

with processes that affect ecosystem productivity and soil microbial processes. Consistent with
our findings, Wiesmeier et al. (2014) reported climatic variables (temperature and precipitation)
as significant controllers of SOC stocks up to 1 m depth in German soils under oceanic climate
(Wiesmeier et al., 2014). Sreenivas et al. (2014) used RF to predict the SOC variability across
semi-arid and humid areas of India in the top 30 cm of soil and found that the top three
environmental controllers were land cover, mean temperature of hottest months, and mean
annual precipitation (Sreenivas et al., 2016). In our analysis, the overall relative importance of
climatic variables was significantly higher than other variables at the global and biome scales.

Soil properties were identified as the second most important controllers of global SOC

stocks. Soil properties impact various processes that govern soil carbon dynamics. For example,
soil properties impact microbial activity, porosity, and oxygen availability in the soil profile,
which directly or indirectly control soil water dynamics, plant growth, and SOC stocks.
Consistent with our findings, Luo et al. (2021) reported that sand content, silt content, and soil
pH were significant controllers of SOC stocks in all soil depths globally.

The Palmer drought severity index, which indicates low soil moisture availability, was a

dominant controller of global SOC stocks. Consistent with our findings, Li et al. (2021) reported
that soil particle size and soil water content were the most influential predictors of SOC variation
(Li et al., 2021). Soil drought, indicating more negative soil water potential and low soil





hydraulic conductivity, can cause tree mortality (Anderegg et al., 2012). Climate extremes like
droughts can impact the structure, composition, and functioning of terrestrial ecosystems and can
thereby severely affect the regional carbon cycle (Frank et al., 2015).

Cation exchange capacity is a soil property that indicates the active soil surface to which

SOC may be adsorbed, and polyvalent metal cations can play a significant role in SOC
stabilization by binding organic compounds to mineral surfaces (O'Brien et al., 2015; Solly et
al., 2020). O'Brien et al., (2015) found that exchangeable soil $Ca^{2+}$ is a significant predictor of
SOC stocks. This relationship is supported by the mechanism that $Ca^{2+}$ and $Mg^{2+}$ promote clay
flocculation and bind organic matter to clay surfaces. Solly et al. (2020) reported that SOC and
cation exchange capacity are significantly related in both topsoil and subsoil with strong positive
relationship.

After climatic factors and cation exchange capacity, topography and vegetation (NPP)

were important controllers of observed global SOC stocks. Effects of NPP on observed SOC
stocks was found to be small (~6% in 0-100 cm soil depth). Similar to our findings, Luo et al.
(2021) reported NPP explaining about 10% of the variation of SOC stocks. NPP delivers the
primary inputs of carbon to soil and NPP generally increases with moisture, temperature, and
$CO_2$ up to a certain limit (Todd-Brown et al., 2013). NPP also depends on the availability of soil
nutrients. Most ESMs overestimate the increase in SOC pools in response to NPP increases
(Todd-Brown et al., 2013). The effects of NPP on SOC also depend on biome type and soil
depths (Luo et al., n.d.; Georgiou et al., 2021). The contribution of NPP on SOC stocks mostly
depends on how much NPP ends up in the soil and how it is translocated to different soil depths.
Georgiou et al. (2021) reported a saturating relationship of SOC stocks with increasing NPP in a





global observational dataset. However, Chen et al., (2018) reported high SOC stocks with
increasing productivity and soil water holding capacity (Chen et al., 2018).

The three CMIP6 ESMs we analyzed predicted SOC stocks mostly as a function of

temperature, precipitation, and NPP. These ESMs simulated positive correlations between SOC
stocks and NPP (Figure 5a), resulting in high SOC stocks in areas with high NPP in most regions
(Shi et al., 2013; Sun et al., 2016). In these ESMs, effects of temperature and precipitation on
SOC stocks are driven by soil respiration. Most current ESMs simulate the response of soil
respiration to temperature using either a $Q_{10}$ or Arrhenius equation (Wynn et al., 2006), such that
a higher temperature causes more soil respiration, and, all else equal, eventually reduces SOC
stocks (Figure 5b). Our results showed diverse control of precipitation on SOC stocks in
different ESMs. Todd-Brown et al. (2013) showed that ESM soil respiration either increases
monotonically with precipitation, or first increases to a plateau under optimal precipitation and
then decreases with further increasing precipitation. Consistent with those results, the ESMs we
analyzed in this study showed different dependence of SOC storage on annual precipitation.

In this study, we found that, in comparison to the patterns that emerged from

observations, ESMs have distinctively different emergent relationships between environmental
factors and SOC stocks. These results could either result from unrealistic parameterization or
missing critical processes in model representation. Our results show that observed global SOC
stocks are controlled not only by temperature, precipitation, and NPP. Effects of other
environmental factors, such as drought severity index and cation exchange capacity should also
be considered in future representations of SOC dynamics in ESMs. It is also imperative to
compare observational data and ESM simulations to improve model structures and
parameterization.





## 5. Conclusion

Our results document disagreement between environmental controllers of SOC stocks in
observations and ESM land models. Specifically, NPP, annual temperature, and annual
precipitation have dominant control in modeled SOC stocks. In contrast, diurnal temperature,
drought index, annual temperature, cation exchange capacity, and other soil related variables are
the dominant controllers of observed SOC stocks. Using field observations and data for
environmental factors, machine learning techniques predict about 60% of the variability in
observed global SOC stocks, while in ESMs, only a few environmental factors predict about
95% of the variability in predicted SOC stocks. Comparisons of derived functional relationships
between the environmental factors and SOC stocks in observations and ESM models also show
discrepancies. These discrepancies indicate the importance of efforts to benchmark ESM land
models and to improve the mechanistic representations that are affected by the observed
dominant environmental controllers. Such an effort could decrease disagreements between
observed and modeled SOC stocks.

**Acknowledgements**
This study was supported jointly by the Laboratory Directed Research and Development
program of Sandia National Laboratories and the Reducing Uncertainties in Biogeochemical
Interactions through Synthesis and Computation Science Focus Area (RUBISCO SFA), which is
sponsored by the Regional and Global Model Analysis (RGMA) activity of the Earth
Environmental Systems Modeling (EESM) Program in the Earth and Environmental Systems



Sciences Division (EESSD) of the Office of Biological and Environmental Research (BER) in
the US Department of Energy Office of Science. Sandia National Laboratories is a multimission
laboratory managed and operated by National Technology and Engineering Solutions of Sandia,
LLC, a wholly owned subsidiary of Honeywell International, Inc., for the U.S. Department of
Energy's National Nuclear Security Administration under contract DE-NA-0003525. Lawrence
Berkeley National Laboratory (LBNL) is managed by the Regents of the University of California
for the U.S. Department of Energy under Contract No. DE-AC02-05CH11231. Oak Ridge
National Laboratory (ORNL) is managed by UT-Battelle, LLC, for the U.S. Department of
Energy under Contract No. DE-AC05-00OR22725.

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



**Figures and Tables**

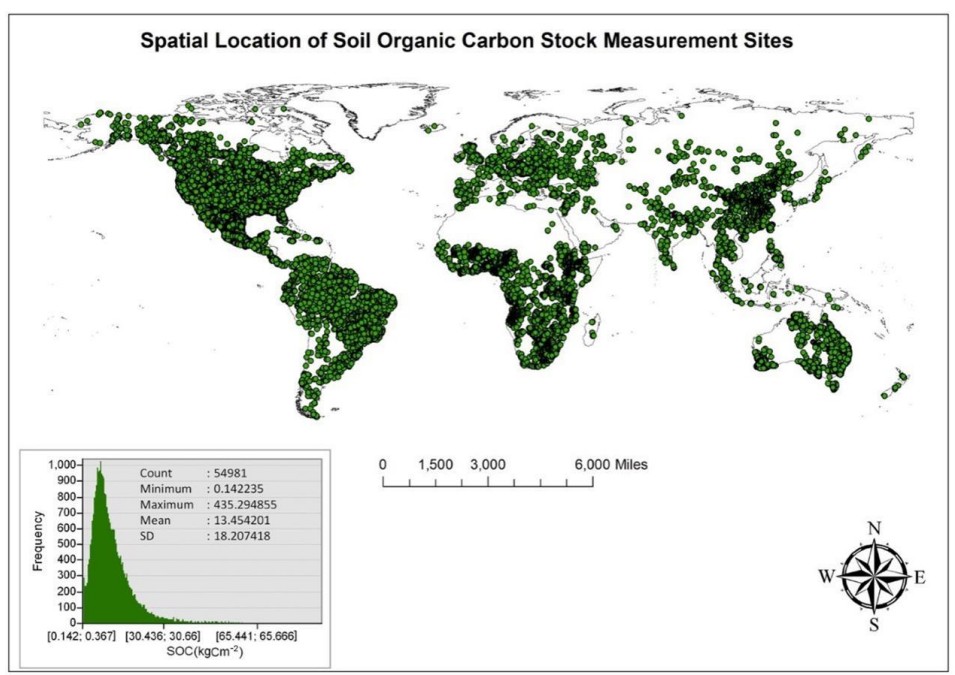

Figure 1. Spatial and statistical distributions of 54,000 soil organic carbon profiles used in this study.

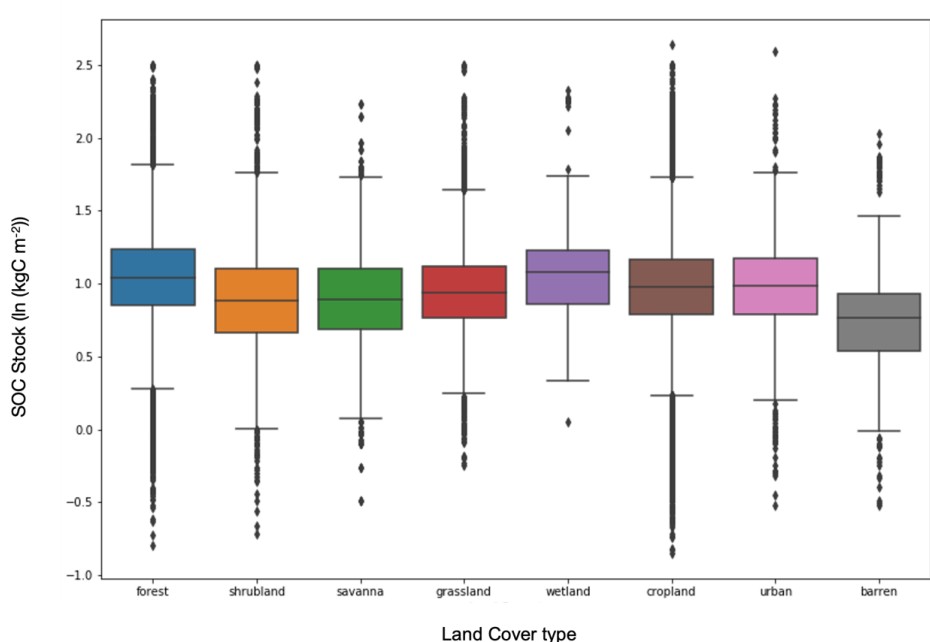

Figure 2: Boxplot of soil organic carbon content (logarithmic scale) for each biome or
land cover type analyzed in this study. The horizontal line in the middle of the boxes is
the median while their lower and upper limits correspond to the first and third quartiles.





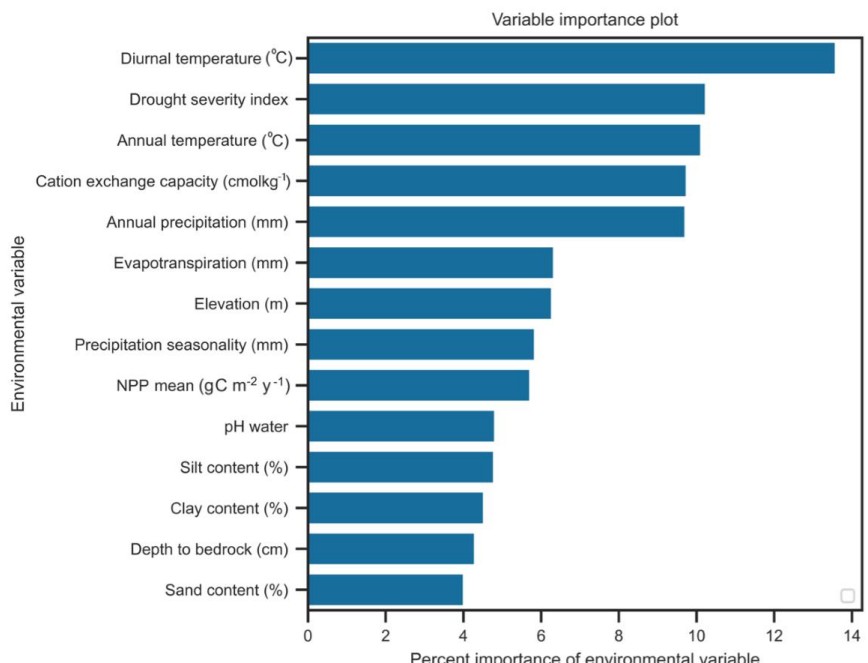

Figure 3: Importance of different environmental factors to predict the global soil organic carbon stocks in observations.





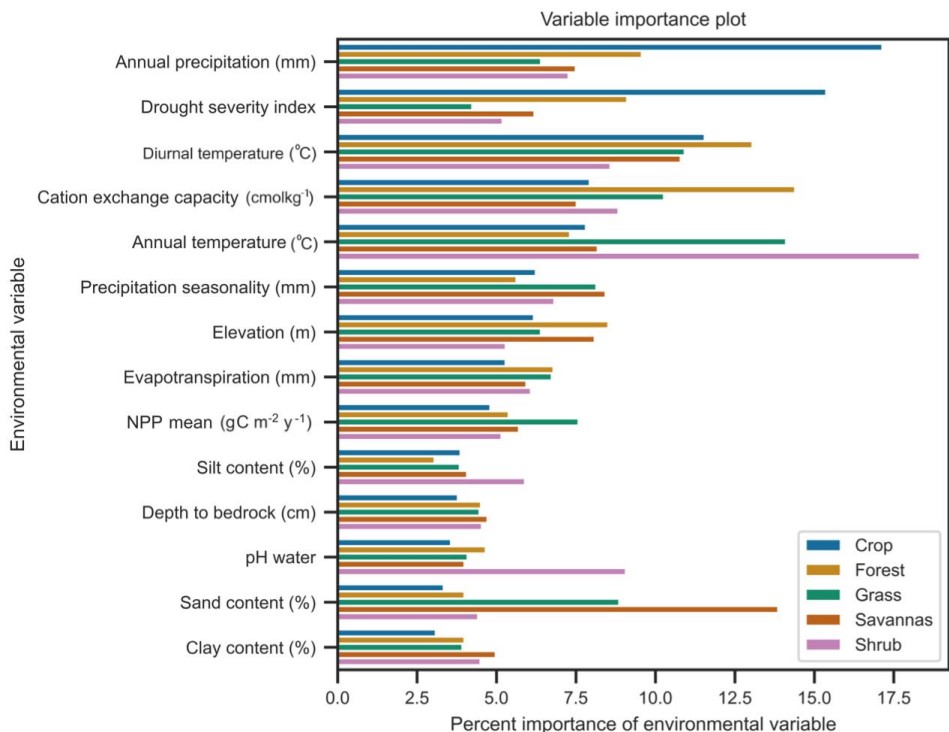

Figure 4: Strengths and importance of environmental controllers of observed SOC stocks within different biomes.




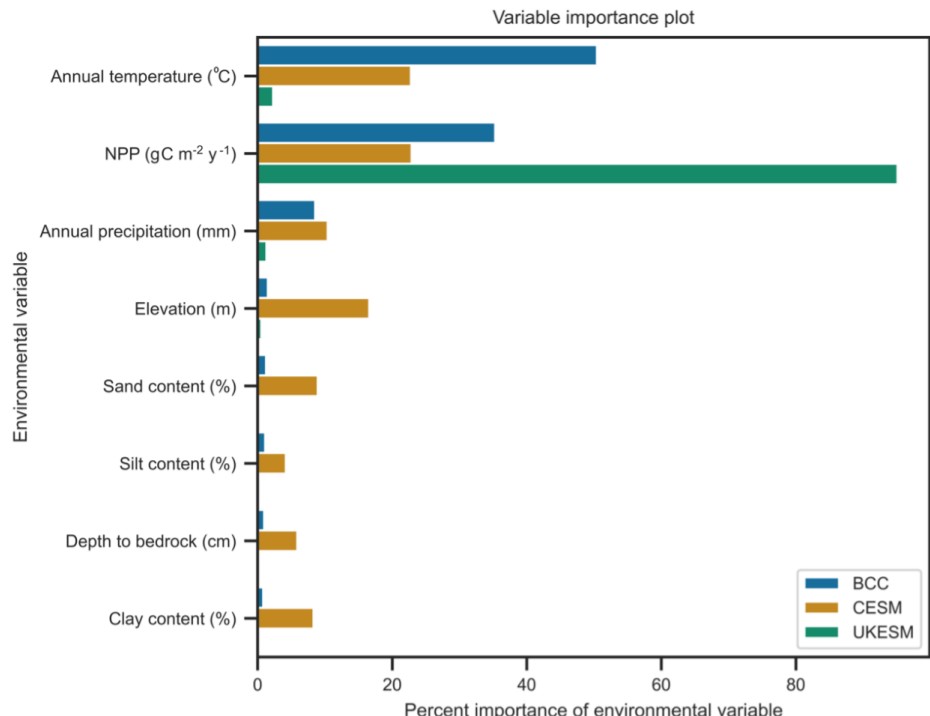

Figure 5: Importance of different environmental factors on global soil organic carbon stocks in three CMIP6 earth system models.





Figure 6: Predictive relationships between environmental factors and soil organic carbon stocks in observations (black line) and CMIP6 earth system models (different colors).





Table 1: Descriptive statistics of global soil organic carbon stocks at 0-100 cm depth interval.

| Location | Depth (cm) | Minimum (kgC m$^{-2}$) | Maximum (kgC m$^{-2}$) | Mean (kgC m$^{-2}$) | Median (kgC m$^{-2}$) | Standard Deviation (kgC m$^{-2}$) |
|---|---|---|---|---|---|---|
| Global | 0-100 | 0.14 | 435.3 | 13.5 | 9.5 | 18.2 |
| Cropland | 0-100 | 0.14 | 435.3 | 12.75 | 9.5 | 16.0 |
| Grassland | 0-100 | 0.56 | 315.9 | 12.1 | 8.7 | 16.8 |
| Forest | 0-100 | 0.16 | 314.4 | 15.9 | 10.9 | 20.7 |
| Shrubland | 0-100 | 0.19 | 312.5 | 13.6 | 7.6 | 25.6 |
| Savannas | 0-100 | 0.32 | 309.1 | 12.6 | 9.2 | 15.2 |





Table 2: Prediction accuracies of Random Forest models across biomes and at global scale in predicting SOC stocks.

| Location | Depth (cm) | R square (RF) | RMSE |
|---|---|---|---|
| Global | 0-100 | 0.61 | 0.46 |
| Cropland | 0-100 | 0.65 | 0.51 |
| Grassland | 0-100 | 0.57 | 0.46 |
| Forest | 0-100 | 0.59 | 0.52 |
| Shrubland | 0-100 | 0.64 | 0.54 |
| Savannas | 0-100 | 0.48 | 0.52 |