# Peer review of "Observational benchmarks inform representation of soil organic carbon dynamics in land surface models"

_Biogeosciences, 2023_

## Author Response (AR1)

Kirsten Thonicke, Associate Editor
Biogeosciences

Dear Dr. Thonicke,

We greatly appreciate the opportunity to revise and resubmit the manuscript. We have carefully considered each of reviewer's comments and found most comments helpful to improve our manuscript. We have done a thorough revision of the entire manuscript, adopted all the relevant suggestions provided by both reviewers, and attempted to eliminate the grammatical errors throughout the manuscript.

Major changes we have made in the manuscript are following;
- We added separate paragraphs in both the Introduction and Discussion sections of the manuscript which provide summaries on two key aspects: (1) the current use of various environmental factors in ESMs and their relation with SOC decomposition in relation to soil moisture, temperature, and microbial activity, and (2) the influence of drought on SOC decomposition, as recommended by Reviewer 1.
- We reclassified the entire SOC datasets using global biomes data (Olson et al., 2001), and re-ran biome level ML analysis and updated Figures 2, 4, & 6 and Tables 1 & 2 as suggested by reviewer 2.
- We have added text to more clearly explain the findings & limitations of this study.

Kind Regards
Umakant Mishra
Principal Member of Technical Staff
Sandia National Laboratories

Please find below our response to specific comments of both reviewers

First, we would like to thank the reviewers for their time and thoughtful critique of this manuscript. We have addressed all the comments and believe that the article has been improved because of the valuable feedbacks. Please find our point-by-point responses below.

**Reviewer 1 comments:**

*The study is interesting because of the general perspective it gives, but it contains also quite some self-evident remarks and some related misunderstandings. It also lacks context, since most of the advancements in modeling in the last 50 years (starting from the temperatures and soil water content relationships with microbial activity) are not considered but you are still making recommendations to modelers. In order to make your statements, you need first to study and briefly review (connecting your results to them) how these processes are represented in ESM, to be honest it sometimes seems you have just a vague idea.*

Response: We thank the reviewer for their thoughtful comments. To address these comments, we have added a separate paragraph in the Introduction section which summarizes how different environmental factors are currently used in ESMs. We also reviewed existing literature on SOC decomposition and its relationship with soil moisture, temperature, and microbial activity as suggested (P4:L67-P5:L92).

*The main of these remarks are all the discussions about introducing in models any variable related to soil moisture or temperature. These relationships are very well known, we know that those are the main factors influencing SOC decomposition and models consider this already quite well.*

Response: We appreciate the reviewers' comment regarding the environmental factors influencing SOC decomposition. However, we feel that the wide range of soil moisture and temperature functions used in models indicates that these relationships are not robustly known. For example, Sierra et al. (2015) Fig 4a and 4c, reproduced below, show the very large differences in temperature and moisture functions used in several common land models. Although these curves reflect the diversity of functional forms used in CMIP6 land models, they are not the same as the relationships we are quantifying from observations, which are better terms as 'emergent functional relationships.

[Figure]

Our results also indicate that ESM land models in CMIP6 do not capture the emergent moisture and temperature controllers on SOC decomposition rates. For example, we found that 3 environmental factors explain more than 96% of variability in ESMs. In contrast, 14 environmental factors only explain 61% of variability in observations. Half of these observationally-inferred environmental factors are edaphic factors which are not adequately represented in ESMs. Among the 3 environmental factors which are major predictors in both ESMs and observations, the emergent functional relationships are very different.

*Using drought instead of precipitations is no improvement. Most SOC models (or maybe all) consider soil moisture simulating it based on precipitations and evapotranspiration (with more or less refined water balance), and have relatively refined response functions of decomposition responding to soil*

*moisture. When the soil moisture falls, the microbial activity in the models (often represented by the kinetics) reduces. This is the main impact of drought on SOC. Most parametric decomposition models go further, representing also a decrease in activity towards the end of the curve, when soil gets close to saturation (this time due to lack of oxygen). You could for example start from the review by Moyano et al., 2013 to have an overview of the discussion about moisture. Concerning temperature effects, you could check first the Lloyd and Taylor 1993, a good citaton classic. But even if temperature is between the two the easier bit, there's still discussion going on (for example Ratkowski). And here we are still just considering first order kinetic models, representing these interactions as external forcing variables as a scaling of the kinetics. There are much more complex models that represent these effects internally to the model itself, for example representing the effect of soil texture on moisture or explicitly considering diffusion of nutrients.*

Response: We thank the reviewer for these helpful suggestions. As suggested, we have reviewed each of the suggested publications and expanded the literature review both in the Introduction and Discussion sections of the manuscript ((P4:L67-P5:L92, P14:L284-293).

*If drought works better in your model than precipitation the main reason (given that you are using models with potentially "infinite degrees of freedom", at least I personally define ML models like that when I use them, being quite illiterate on the topic) is probably that you are not including evapotranspiration in your model (while most/all ESM will probably do calculate the soil water balance) while drought contains, in some sense, information about that too.*

Response: In our study, we tried to include a wide range of environmental factors that are available and can be related to SOC dynamics. The Palmer drought index that we used in this study explicitly includes evapotranspiration, precipitation, and temperature. Therefore, as you mentioned, the drought index includes controls on the soil water balance, making it a better predictor than precipitation alone. In the revised manuscript, we have discussed its importance in predicting SOC (P14:L283-294).

*Also your speculation about the causes of why drought was so important in your model (like 262-264) are not so convincing. For sure it might be that there is also an effect on inputs, but those should be considered in your model(s) already by using NPP. While it is very well known (and how much, numerically) that soil moisture affects microbial activity.*

Response: We agree with the reviewer that the text in L262-264 can be improved and made more focused. As suggested we have modified that text in the revised manuscript (P14: L285-294).

*Summarizing, you are probably asking the wrong questions to your model(s). Saying that temperature related and moisture related indicators (whatever those are, since your model has "infinite" degrees of freedom), is extremely self evident since half a century. While asking to your model how much would it matter to include also some edaphic parameter, and which one, one a global scale for predictions, that would be an interesting question to read about.*

Response: While we agree that temperature and moisture are very well known to affect SOC decomposition rates, their emergent functional relationships are not well known and vary strongly

between land models (e.g., Sierra et al. 2015). Therefore, the objective of our study was to benchmark environmental control representations in the current generation of ESMs using existing observations and ML approaches. The study that we conducted clearly showed (1) dominant environmental controllers of SOC stocks at global scale in observations and CMIP6 ESMs, and (2) the mathematical relationships between the dominant environmental controllers and SOC stocks in both observations and ESMs. To address this concern, we have added text to the revised manuscript discussing the known importance of soil moisture and temperature on SOC stocks, and highlighting the wide range of functional forms included in CMIP6 land models. We also attempted to address the reviewer's comment about how edaphic parameters inform global-scale SOM predictions. Out of 14 dominant environmental factors that our ML models selected as global predictors of SOC stocks in observations, 7 were edaphic factors. In the three CMIP6 ESMs we evaluated, only CESM used 5 of these edaphic factors. Interestingly, the cation exchange capacity, which is the most dominant environmental factor inferred from observations, is not used in any CMIP6 ESMs that we evaluated. We have modified the text in the Discussion section to highlight these findings as the reviewer suggested (P16:L329-345).

*Line 293-294: you don't need this kind of study to demonstrate different controls of moisture for different ESM. You can simply read which functions they rely upon, if those functions are different (they are) then the controls will be different. I think you should study the main functions available for that, and which function has been implemented in which model.*

Response: To clarify, our objective was to benchmark how the existing environmental controls are related to the emergent SOC stocks in ESMs and observations. These emergent relationships may differ from the relationships coded in land models for several reasons, including interactions between multiple stressors (e.g., nutrients, moisture, temperature, light), time scales of analysis, and model differences in calculating moisture and temperature). We have added text to the revised manuscript to clarify these points. Our results show varying influences of different variables on SOC stocks across different ESMs. In addition to the dominant environmental controllers of SOC stocks, we also report observationally-inferred relationships between dominant variables and SOC stocks. As suggested by the reviewer in this and earlier comments, we have also modified the text in the Introduction section of the manuscript to include existing functions that ESMs currently include to represent control of these environmental factors (P4:L67-P5:L92).

*Your conclusions seem off track. Line 310-311: I would say that there's no disagreement, all SOC models are using temperature and moisture of the microbial environment to control decomposition, those processes are well taken care of (better than using drought alone). Different models will of course rely on different variables to represent the same processes, the fact yours relies on diurnal temperature instead of soil temperature or daily temperature does not allow you to make inferences on other models, it depends on the functions they use. But they all agree that we need to represent the impact that the water present in the microbial micro-environment has on kinetics, and the effect that temperature in such micro-environment has also on the kinetics.*

Response: As mentioned above, we agree with the reviewer that it is well known that soil moisture and temperature are important controllers of SOM decomposition. To clarify this point, we have

added a sentence to this effect in the revised manuscript. However, our results demonstrate that the model and observations have very different inferences of the emergent functional form of these relationships, and importantly, the number of controllers that need to be considered. The models dramatically underestimate the number (3 versus 16) and type (e.g., edaphic) of important controllers that we inferred from observations. We note that a wealth of literature exists documenting discrepancies between ESM land models and observed SOC stocks and dynamics. Consistent with our study, multiple previous studies (Collier et al., 2018; Georgiou et al., 2021; Luo et al., 2012; Todd-Brown et al., 2013; document the need for model benchmarking studies to identify discrepancies and improve model structures to reduce uncertainties in predicting carbon climate feedbacks (P17:L353-363).

*So, concluding, the study could have some potential but it requires a much better and extensive work on documenting the state of the art in detail. Understanding how the problems you talk about are already dealt with in models (and I mean at the level of the single functions) will also help you to repurpose your conclusions.*

Response: Thank you for indicating the merits of our study. As suggested by the reviewer, we have modified the manuscript text in multiple places, added new references, and provided greater details about the environmental controllers that are represented in models.

*I also suggest you to shift your focus a bit, you are probably having a bit too ambitious goals (of making a big impact on modeling). Your approach is interesting for me (I am a modeler myself) because it offers insights on processes that ok, we know well in principle, but still they vary in different environments, there might be interactions with environmental factors changing the relationships, and so on. The global perspective of your study is interesting already, even in case you won't revolutionize anything.*

Response: Thank you for your comments. Our goals are actually quite clear: to document existing differences between global observations and ESMs regarding: (1) dominant environmental controllers of SOC stocks, and (2) the emergent relationships between the dominant environmental controllers and SOC stocks.

*I have minor (but still important) concerns about validation too. How did you trained your RF models? Can you ensure that the validation is completely independent? For example if you used Caret to train the metaparameters, you might have a spillover of the training in validation (because you select the metaparameters with the crossvalidation results). Another big issue would be to ensure that the data points of each fold of the crossvalidation are not correlated with any data point in the training. For example if you have more propfile from one single site, some would end in validation some in training (for each fold), injecting information from validation into training. If you are selecting instead at the site level (or if it corresponds to the data point level) it's all fine.*

Response: Thank you for the comments, the model was trained (70% of the data) and tested using the other 30% of data. We have not used multiple profiles from a single site since each site contained a single soil profile, so there was no spillover effect. For splitting data into calibration and validation

datasets, we have used a standard procedure that we have used in our other studies to split the calibration and validation datasets in a spatially balanced way (Mishra et al., 2022; Mishra et al., 2021; Mishra et al. 2020).

*Concerning the GAM, how did you validate them? You say you used Rˆ2, but based on which dataset did you calculate it?*

Response: A similar model validation approach was used for both RF and GAM approaches. 70% of the data was used for training the model and 30% of the data were used for model testing. We have modified the text in the revised manuscript to more clearly explain how model calibration and validation were done.

*In general, please be extremely specific about your validation approaches, in particular discussing why you believe there is no spillover of information between training and validation and why the two are supposed independent.*

Response: Thank for your suggestions. As described above, the model was trained and tested using independent data sets, i.e., 70% of the data was used for training and 30% of data was used for testing. Each site contained a single soil profile, so there are no spillover effects. For splitting that data into calibration and validation datasets, we have used standard procedures that we have used in our previous studies to split the calibration and validation datasets into spatially balanced way (Mishra et al. 2020; Mishra et al., 2021; Mishra et al., 2022).

*You also need to describe better the study on the ESM data. What are the SOC data you mention on line 156? Are those simulated data or measured?*

Response: Thank you for your comments. We have added text in the Methods section of the revised manuscript and provided greater details about both field observations and CMIP6 ESM data that we used in this study (P10:L187).

References:

Collier, N., Hoffman, F. M., Lawrence, D. M., Keppel-Aleks, G., Koven, C. D., Riley, W. J., Mu, M., and Randerson, J. T.: The International Land Model Benchmarking (ILAMB) system: design, theory, and implementation, Journal of Advances in Modeling Earth Systems, 10, 2731–2754, , https://doi.org/10.1029/2018MS001354, 2018.

Georgiou, K., Malhotra, A., Wieder, W. R., Ennis, J. H., Hartman, M. D., Sulman, B. N., Berhe, A. A., Grandy, A. S., Kyker-Snowman, E., Lajtha, K., Moore, J. A. M., Pierson, D., and Jackson, R. B.: Divergent controls of soil organic carbon between observations and process-based models, Biogeochemistry, 156, 5–17, https://doi.org/10.1007/S10533-021-00819-2, 2021.

Luo, Y. Q., et al.: A framework for benchmarking land models, Biogeosciences, 9, 1899–1944, https://doi.org/10.5194/bgd-9-1899-2012, 2012.

Mishra, U., Yeo K., Adhikari, A., Riley, W.J., Hoffman, F., Hudson, C., and S. Gautam, S.: Empirical relationships between environmental factors and soil organic carbon produce comparable prediction accuracy as the machine learning, Soil Science Society of America Journal, 86, 1611-1624, doi:10.1002/saj2.20453, 2022.

Mishra, U., et al.: Spatial heterogeneity and environmental predictors of permafrost region soil organic carbon stocks. Science Advances,7, eaaz5236, doi: 10.1126/sciadv.aaz5236, 2021.

Mishra, U., Gautam, S., Riley, W.J., and F. Hoffman, F.: Ensemble machine learning approach better predicts the spatial heterogeneity of surface soil organic carbon stocks in data-limited northern circumpolar region, Frontiers in Big data, 3, 40, doi: 10.3389/fdata.2020.528441, 2020.

Sierra, C. A., Trumbore, S. E., Davidson, E. A., Vicca, S., & Janssens, I. (2015). Sensitivity of decomposition rates of soil organic matter with respect to simultaneous changes in temperature and moisture. Journal of Advances in Modeling Earth Systems, 7(1), 335-356.

Todd-Brown, K. E. O., Randerson, J. T., Post, W. M., Hoffman, F. M., Tarnocai, C., Schuur, E. A. G., and Allison, S. D.: Causes of variation in soil carbon simulations from CMIP5 Earth system models and comparison with observations, Biogeosciences, 10, 1717–1736, https://doi.org/10.5194/BG-10-1717-2013, 2013.

**Reviewer 2 comments:**

Review for Observational benchmarks inform representation of soil organic carbon dynamics in land surface models, bg-2023-50

General comments

*The paper is well structured, easy to follow and interesting. The subject is very important and the results of this paper can be useful for modelling of climate change and carbon cycles. It shows that some processes, that are often omitted in the process modelling of the gcms have an important influence on the soil C stocks, and that the physical properties of the soils have a larger influence of soil C stocks in observations than in the studied models.*

Response: We sincerely appreciate the thoughtful and encouraging comments of the reviewer.

Specific comments

*Lines 85-86. The upper meter of soil isn't necessary the whole profile. Topsoil also isn't always 30 cm thick, but the thickness depends on the soil formation at the soil profile. It would be better just use something like "upper 30 cm of soil" and "upper meter of soil" and not use already existing names. Also, don't you underestimate the carbon stock of wetlands if you only count the uppermost meter or 30 cm, when many peat lands have a larger depth of peat?*

Response: We thank the reviewer for these suggestions. As suggested, in the revised manuscript, we have modified the text used to describe depth descriptions. We agree with the reviewer that the depth descriptions of 30 cm and 1 m will not account for the total peatland SOC stocks as peatlands store more carbon to a much greater depth. But, as we have included soil samples from all kind of soils, we used depth descriptions of 30 cm and 1 m which are often used in literature (P6:L113-114).

*Line 118. There are no factors that cover biotic factors like presence or non-presence of certain species that affect soil structure and soil carbon more than others (spruce trees and earthworms for example) and no factors that cover details in management by humans, like prescribed burnings, use of organic or inorganic fertiliser or no fertiliser, irrigation of agricultural soils, presence or no presence of water draining ditches in forests and agricultural land, whole tree removal on clear cuts in forests or removal of only stems, with tops and branches left on site, forests or shrublands used heavily for fire wood collection that removes most dead wood or not. Soil moisture is only included by a drought index, which, I assume, does not capture the average soil moisture but the drier extremes. Nitrogen availability is also not included, even though nitrogen affect decomposition of organic matter and NPP.*

Response: We agree with the reviewer that the SOC dynamics of natural and managed ecosystems are different. We also agree that the results may have been different if different environmental predictors have been used. We attempted to conduct a global model benchmarking study such that the findings can inform current generation of ESMs. In this study, our specific objectives were to (1) identify the dominant environmental controllers of SOC stocks at global scale both in observations and CMIP6 ESMs, and (2) derive and compare the mathematical relationships between the dominant environmental controllers and SOC stocks in both observations and ESMs. To meet these objectives, we used 46 environmental factors which covers all the environmental factors that have been used in the current generation of CMIP6 ESMs.

To appropriately address the reviewer's concerns we have added a separate paragraph at the end of the Discussion section in the revised manuscript explaining the limitations of our approach. In particular, we mentioned that ecosystem specific (for example croplands and forests) environmental factors should be used in future studies as they may improve the SOC prediction accuracy in observations (P17:L346-350).

*Line 180 ff. Is it reasonable to divide land uses in so few classes? The averages in soil C stock is close to each other between the different classes you use? Would other, finer, classes yield similar averages, or would there be forest classes with much higher and much lower average C stocks than the average of all forest? Would the main driving factors be the same for all forest subclasses as they are for the one forest class you are using, or would they be different for e.g. tropical forest and temperate deciduous forest etc. And the same for subclasses of barren land and your other land classes, would the subclasses react in the same way as the large class, even though they might be very different (barren land can be barren for very different reasons for example, and urban land can be mostly concrete and asphalt or mostly gardens, depending on population density and other factors)?*

Response: We agree that the dominant environmental controllers and predictive power of ML models will differ if the SOC stock observations were divided using different land cover categories. But, the ML approach is a data intensive approach and requires a large number of data points to produce stable results. That is why we divided the entire datasets into eight different land cover classes. In order to better address the reviewer's concerns, we have re-categorized our datasets using global biomes (Olson et al., 2001). (P25: Figure 2, P27: Figure 4, P30: Table 1, P31: Table 2).

*Discussion: Discuss the implications of your model being a statistical model, whereas the gcms are far more complex process-based models – how can they incorporate your findings? A drought index, for example, is only an index describing the result of the interaction between temperature, precipitation, evaporation and other processes that are already in the gcms, and it will not be constant when climate is changing – what ways could there be to incorporate it? CEC is also changing with time and soil pH, soil carbon content et c, and must in a process-based model be modelled, together with the important processes related to soil pH, such as base cation concentrations.*

Response: Thanks for these insightful comments. In response to these comments, we have modified the text in the Discussion section of the manuscript and suggested ways that edaphic controls, which are half of the dominant controllers of SOC in observations, can be incorporated in ESMs. We also mentioned that our mathematical relationships can be used to benchmark ESM results. While our results can not directly be used to develop model parameterizations, they can: (1) point to categories of functional forms for controllers; (2) inform where effort may best applied to improve model functional forms (e.g., to the dominant controllers); and (3) inform modelers of where their model may have very different functional forms for emergent relationships than exist in the observations (P16:L339-P17:345).

*Line 288. There is no Figure 5a, it is Figure 6a.*

Response: Thanks for indicating the error; the figure number is corrected in the revised manuscript (P15:L318).

*Line 293. There is no Figure 5b. Figure 6e is the figure with temperature effect, but I don't understand what you mean by eventually reduce soil carbon – the curve is falling already at low temperatures.*

Response: Thanks for indicating the error, the manuscript text has been modified to correct the figure number and the observed and modeled trends of changes in SOC with increase in temperature (P16:L328).

*Line 297. Refer to Figure 6b?*

Response: Thank you for finding the error, the figure number is corrected in the revised manuscript (P16:L328).

*Figure 6. Some measure of the spread around the average of the observed values would be very interesting. Also, it should be the same y-axis scale on all figures a-f. Figure 6f: The curve of the observed values is rather flat for 0 - 2000 m above sea level and relatively little land has an elevation higher than that. Have you made sure that the data from higher elevations don't have a disproportionate effect on your model? And that the effect of high elevation on soil carbon isn't only an effect of exposure to erosion?*

Response: Thank you for these suggestions. We have made the scale of the Y-axis the same in all figures. We had 1543 samples from elevation greater than 2000m, which is about 2.8% total samples. We have reanalyzed the relationships between elevation and SOC and updated the figures and text in the revised manuscript (P29: Figure 6).

Reference:

Olson, D. M. et al. Terrestrial Ecoregions of the World: A New Map of Life on Earth: A new global map of terrestrial ecoregions provides an innovative tool for conserving biodiversity. Bioscience 51, 933-938, doi:10.1641/0006-3568(2001).